# A human cadaveric model for venous air embolism detection tool development

**Nathaniel L. Robinson**[1], **Chris Marcellino**[2,3]*, **Matthew Johnston**[2], **Arnoley S. Abcejo**[2]

1 Department of Cardiovascular Surgery, Mayo Clinic, Rochester, MN, United States of America, 2 Department of Anesthesia, Mayo Clinic, Rochester, MN, United States of America, 3 Department of Neurologic Surgery, Mayo Clinic Health System Eau Claire, Eau Claire, WI, United States of America

* marcellino.christopher@mayo.edu

**Data Availability Statement:** All data are in the manuscript and/or supporting information files.

**Funding:** This work was supported by the Mayo Clinic Anesthesia Small Grant funding, which is made possible by the Mayo Clinic Center for

## Abstract

### Purpose

A human cadaveric model combining standard lung protective mechanical ventilation and modified cardiac bypass techniques was developed to allow investigation into automated modes of detection of venous air emboli (VAE) prior to in vivo human or animal investigations.

### Methods

In this study, in order to create an artificial cardiopulmonary circuit in a cadaver that could mimic VAE physiology, the direction of flow was reversed from conventional cardiac bypass. Normal saline was circulated in isolation through the heart and lungs as opposed to the peripheral organs by placing the venous cannula into the aorta and the arterial cannula into the inferior vena cava with selective ligation of other vessels.

### Results

Mechanical ventilation and this reversed cardiac bypass scheme allowed preliminary detection of VAE independently but not in concert in our current simulation scheme due to pulmonary edema in the cadaver. A limited dissection approach was used initially followed by a radical exposure of the great vessels, and both proved feasible in terms of air signal detection. We used electrical impendence as a preliminary tool to validate detection in this cadaveric model however we theorize that it would work for echocardiographic, intravenous ultrasound or other novel modalities as well.

### Conclusion

A cadaveric model allows monitoring technology development with reduced use of animal and conventional human testing.

Translational Science Activities (CTSA) through grant number UL1TR002377 from the National Center for Advancing Translational Sciences (NCATS), a component of the National Institutes of Health (NIH). The funders had no role in study design, data collection and analysis, decision to publish, or preparation of the manuscript.

**Competing interests:** The authors have declared that no competing interests exist.

## Introduction

Despite the detection capability of transesophageal echocardiogram (TEE) [1, 2] and precordial Doppler ultrasonography, venous air embolism (VAE) remains one of the leading neurosurgical complications [3]. This surgical complication is well described in seated neurosurgery and head and neck [4] surgery, but can also been seen in prone [5] surgical cases and any operation where the surgical wound is above the phlebostatic axis. Both intraoperative TEE and precordial Doppler can be logistically demanding and have significant technical limitations in terms of placement and ease of use [6]. TEE is sensitive [7] but is not suitable to continuous monitoring. While precordial Doppler is relatively less sensitive and specific and can be difficult to perform in many patients and in certain surgical positions (e.g., prone). End-tidal CO2 changes are well described but non-specific, and do not correlate well with recovery after nor severity of VAE [8, 9]. TEE also has rare, yet serious risks including viscus perforation and its placement is relatively contraindicated in certain patient populations.

Perhaps most significantly, neither method has been significantly adapted to automated notification (i.e., alarming) or central monitoring. Many, if not most cases of significant morbidity and mortality from VAE involved detection of the air entrainment only after control of hemodynamics has been lost. Development of a continuous monitoring tool that is easy to apply and use, does not require advanced training, and is inexpensive would be advantageous and potentially improve surgical safety and quality in certain cases.

An experimental model for the investigation of new devices or modifications to existing detection modalities is needed to help further this goal. Cost, ethical concerns and increasing scrutiny of animal research suggests that a cadaveric model may be ideal in preliminary investigation or testing. Such a model would be useful for both ultrasound and impedance based detection schemes, however we used an impedance based approach to demonstrate air entrainment. We, therefore, describe an experimental cadaveric apparatus useful for development of VAE detection modalities.

## Methods

A fresh frozen cadaver including head, neck, chest, and abdomen and pelvis was obtained via our hospital's anatomical bequest program (Mayo Clinic Department of Anatomy, Rochester, Minnesota, USA), which had previously been exsanguinated but not chemically preserved. The individual had bequeathed his body for the purposes of medical research and education at our institution. After our experimentation, which took place from February 22nd to 24th, 2023, the specimen was used for educational dissection by medical students and other surgical teams at our institution and was ultimately cremated and honored in an annual ceremony performed by our anatomical lab with the families of the recent donors. Additionally, the experimentation was approved by the Mayo Clinic Institutional Review Board Biospecimens Subcommittee under study ID 22-008554, who did not require explicit written consent given the nature of the anatomical bequest.

The specimen was originally approximately 80 kg and 180 cm in height, which yielded an approximately 50 kg cadaveric specimen. In order to simulate cardiac and pulmonary blood flow, a modification was employed to reverse the normal flow in cardiac bypass (Fig 1), whereby venous return cannula were placed in the either the abdominal or the distal thoracic ascending aorta and arterial supply cannula in the inferior vena cava (IVC). In comparison, standard cardiopulmonary bypass replaces the heart's pumping function with forward arterial flow moving through a cannula in the ascending aorta and venous flow exiting the body from cannulas in the right atrium or superior and IVC. This configuration allows the heart to be stopped to permit cardiovascular surgery. Instead of bypassing the heart, we sought to utilize

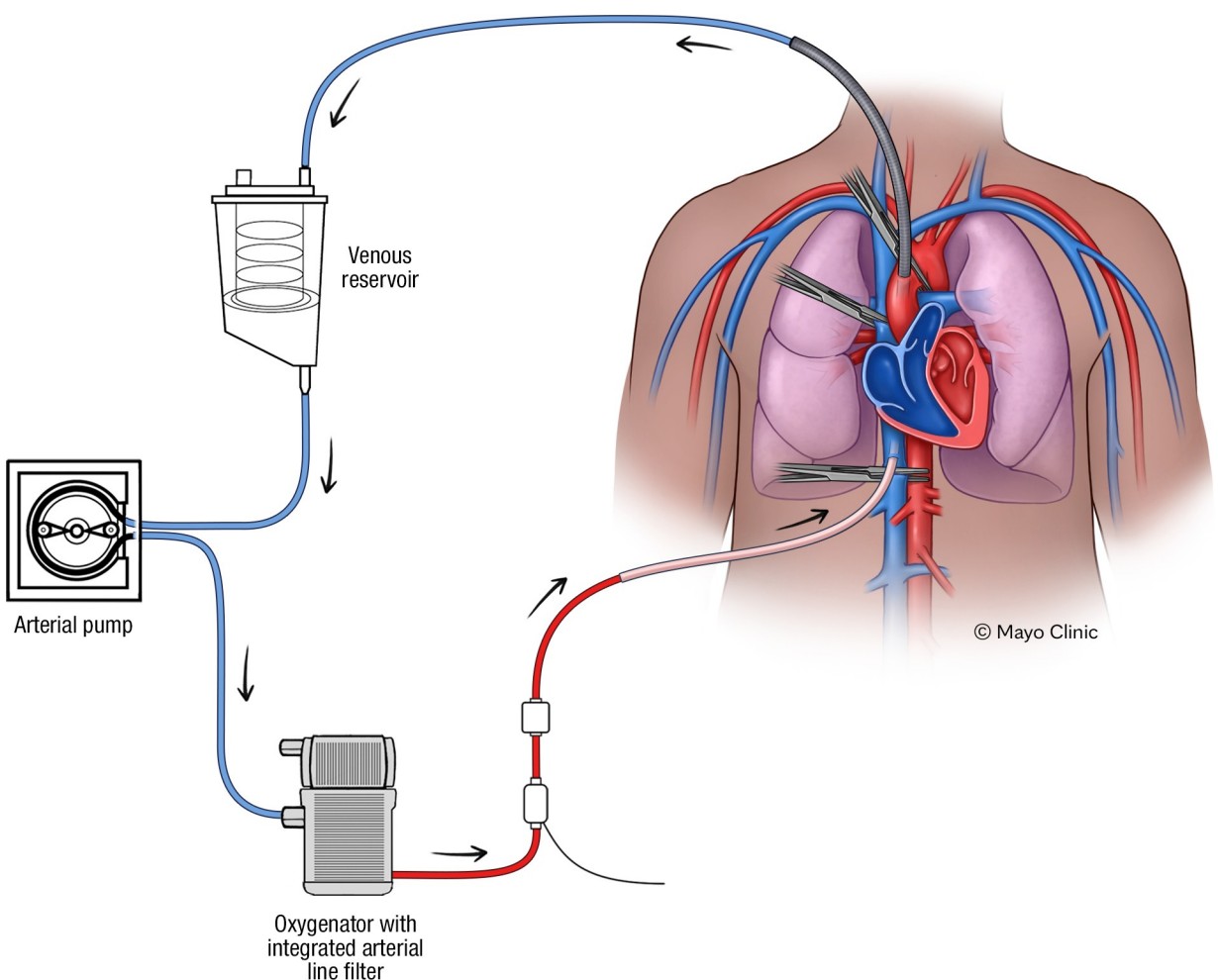

**Fig 1. Schematic diagram of reversed cardiac bypass cadaveric cardiopulmonary perfusion technique.** An arterial cannula is inserted into the vena cava to allow blood flow to the heart and lungs preferentially, followed by drainage into a venous cannula in the distal ascending aorta.

the external pump to allow selective perfusion of the heart and lungs with relative exclusion of the peripheral organs, thus mimicking forward cardiac flow through the right and left heart and pulmonary vasculature in order to allow detection of entrained air.

All major extremity vessels were ligated to prevent loss of intravascular volume after initiation of flow. In the upper extremity, this included bilateral brachial arteries and veins as well as cephalic and basilic veins. In the lower extremity this included bilateral femoral arteries and veins. We then performed a midline abdominal incision from xiphoid to pubic symphysis followed by bilateral subcostal extension. An umbilical hernia with mesh was noted on abdominal entry but other than this, no signs of surgical intervention were noted. The specimen had a fully intact diaphragm and esophagus with a small type 1 paraesophageal hernia. There were grossly normal stomach, spleen, liver, gallbladder, greater omentum, small intestine and colon. In serial, we then preformed 3 dissection techniques, moving from two decreasingly peripheral cannulation approaches to last an intrathoracic approach via thoracotomy.

In order to simulate respiratory variation, the cadaver was intubated with an 8.0 mm endotracheal tube using direct laryngoscopy and connected to the Hamilton G5 mechanical ventilator (Hamilton Medical, Bonaduz, Switzerland), the ventilator was set to Continuous

Mandatory Ventilation (CMV) mode ventilation with a tidal volume of 500 mL (approx. 6.8 mL/kg IBW), a rate of 15 breaths per minute, positive end expiratory pressure (PEEP) of 5 cmH2O and FiO2 of 21%. Prior to initiating flow we were able to ventilate with nominal peak pressures of 21 cmH2O with minimal air leak.

## Exposure techniques

In the initial first extrathoracic approach, colon and small intestine were retracted cephalad, and the rectum was retracted laterally. The midline inframesocolic space and the inferior border of the root of the mesentery was identified and the bifurcation of the abdominal aorta was palpable as it approached the sacral promontory. The retroperitoneal fat was dissected down to the level of the aorta and IVC. The dissection was carried inferiorly exposing the bifurcation and iliac arteries and veins. These vessels were ligated with 0 silk sutures. The dissection was then carried cephalad exposing the great vessels up to the level of the crossing of the renal vein. The inferior mesenteric artery and vein were noted at this point and ligated. A double purse string technique with 3-0 Prolene suture was used on both the aorta and IVC just superior to the bifurcation. Incisions in the great vessels were performed and the cannulas were placed (see Fig 2) A 20 French aortic cannula EZ glide (Edwards Lifesciences, Irvine, California,

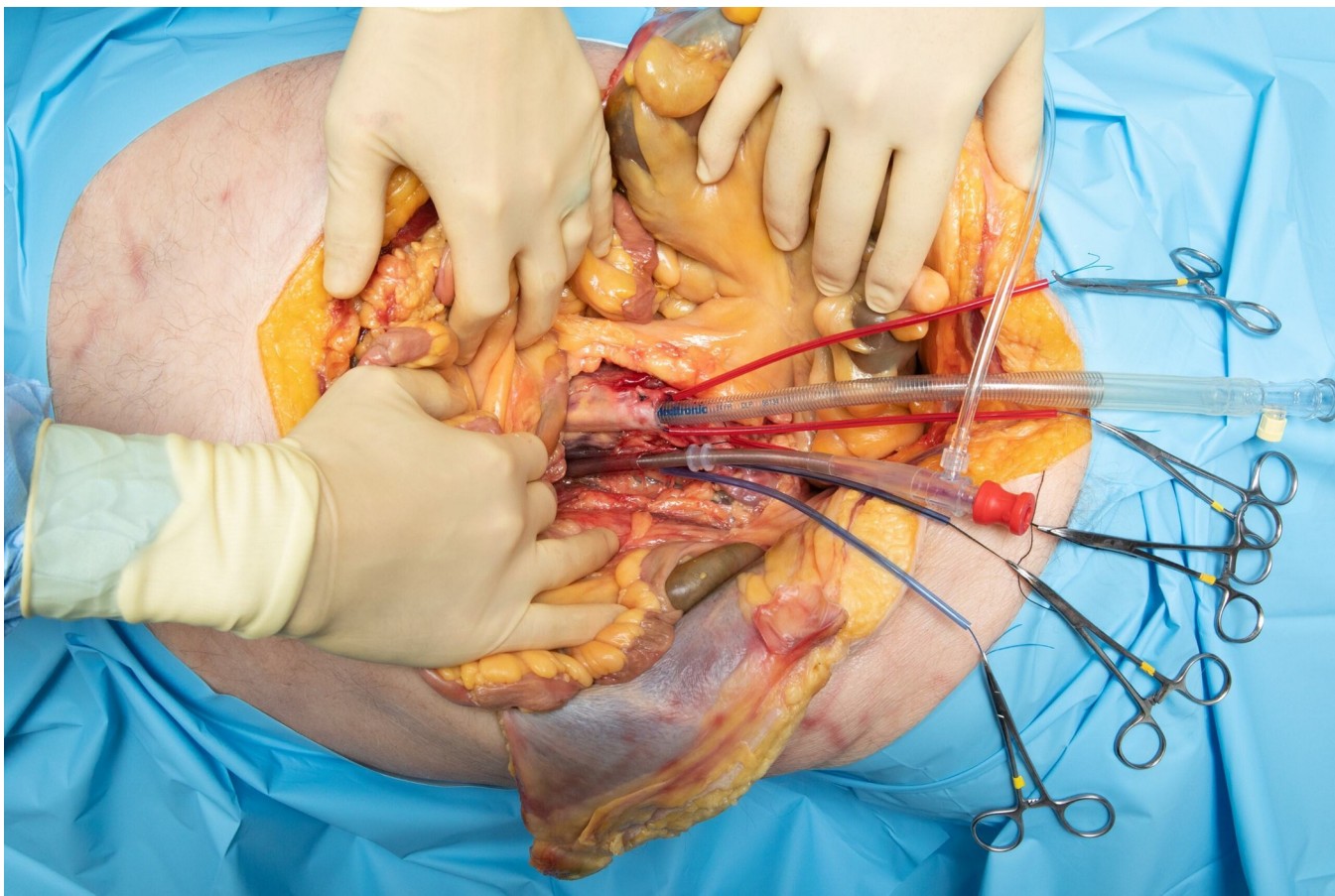

**Fig 2. First approach with distal reverse cannulation.** Initial exposure with reverse (heart-excluding) bypass cannulation of the abdominal aorta and IVC, using a double purse string technique with 3-0 Prolene at the level of the iliac bifurcation. The inferior mesenteric artery and vein were noted at this point and ligated. The arterial cannulae was placed in the aorta, and venous cannulae was placed in the IVC to simulate cardiac and pulmonary flows at the exclusion of the systemic circulation, which is reversed from what is normally done in bypass.

USA) and 32 French bullet venous cannula (Medtronic Inc., Minneapolis, Minnesota, USA) were selected and the arterial cannula was placed in the IVC, and the venous cannula was placed in the abdominal aorta. The cannulae were connected to normal saline primed 9.5 mm (3/8 in.) tubing which was connected to the CentriMag centrifugal pump (Abbott Laboratories, Abbott Park, Illinois, USA.) After clearance of intravascular debris, and switching to a roller pump we achieved a pump flow of 1.56 L/min, at 249 RPM, with 52 mmHg of positive pressure and -32 mmHg of negative pressure under vacuum assist with continuous normal saline infusion (see S1 Video).

After approximately 10 minutes of bypass, mechanical ventilation pressures increased which necessitated suctioning of fluid in the endotracheal tube. We then clamped the endotracheal tube to limit fluid egress as we hypothesized that the little of oncotic pressure from normal saline circulation alone in conjunction with tissue destruction from high flow rates allowed rapid flow of fluid transvascularly. We also observed significant periorbital and facial edema, skin edema in the chest and abdomen, visceral edema with a significant increase in size and volume of the liver, stomach, spleen and especially small intestine and colon. We also noted that at this time 15 L of fluid had been lost from circuit based on input volumes into the patient, which was consistent with observed end organ engorgement via venous-to-capillary bed extravasation with reduced return to the bypass circuit.

In our second approach, we undertook more central but still extrathoracic cannulation to reduce volume losses to the periphery with improvement in the qualitative degree of volume loss. We started by eviscerating the abdomen (Fig 3). A right and left medial visceral rotation were performed and the entire colon was mobilized. The root of the mesentery was dissected, and the superior mesenteric artery and vein were identified and ligated. A Kocher maneuver exposed duodenum. The rectum and third part of the duodenum were ligated and the intestines were removed from the abdomen. The right and left triangular ligaments of the liver were mobilized to the hepatic veins which were ligated and portal structures were identified. The splenic attachments were mobilized, and the esophagus was circumferentially dissected to the level of the superior abdominal aorta. The celiac axis was identified from the left and further dissected through the lesser sac and ligated. The liver, spleen, pancreas, stomach, and omentum were removed from the abdomen leaving retroperitoneal structures including the great vessels, kidneys, adrenals, and stump of the esophagus. At this point, double purse string sutures with 2-0 Prolene were placed around the right left and middle hepatic veins in the IVC and the 20 French forward flow cannula was once again inserted and secured with a Rumel tourniquet with the tip of the cannula in the right atrium. The inferior aspect of the IVC was dissected circumferentially and ligated with a 0 silk suture to prevent reverse flow. The 32 French receiving cannula was left in its insertion point, but advanced further into the descending thoracic aorta and once again secured with a Rumel tourniquet. We again established flow at 1.5 L/min. at the previous settings now in the absence of mechanical ventilation but again with considerable volume losses.

A third exposure was completed to attempt to eliminate volume loss into the superior vena cava (SVC). A midline sternotomy was performed and a sternal retractor was placed. The pericardium was opened exposing the aortic root and pulmonary artery down to the level of the diaphragm (Fig 4). The IVC and aortic cannulas were palpated and found to be in acceptable position. The right and left pleural space were opened revealing multiple liters of extravascular fluid which had extravasated during the previous approaches. To direct the flow of air and fluid through the IVC into the right atrium and into the right ventricle and pulmonary artery (which is the normal path of a VAE), the SVC and IVC (around the instillation cannula) were dissected and snared with Rumel tourniquets which blocked the SVC and retrograde flow back down the IVC pushing all blood into the right atrium. We then moved the venous return

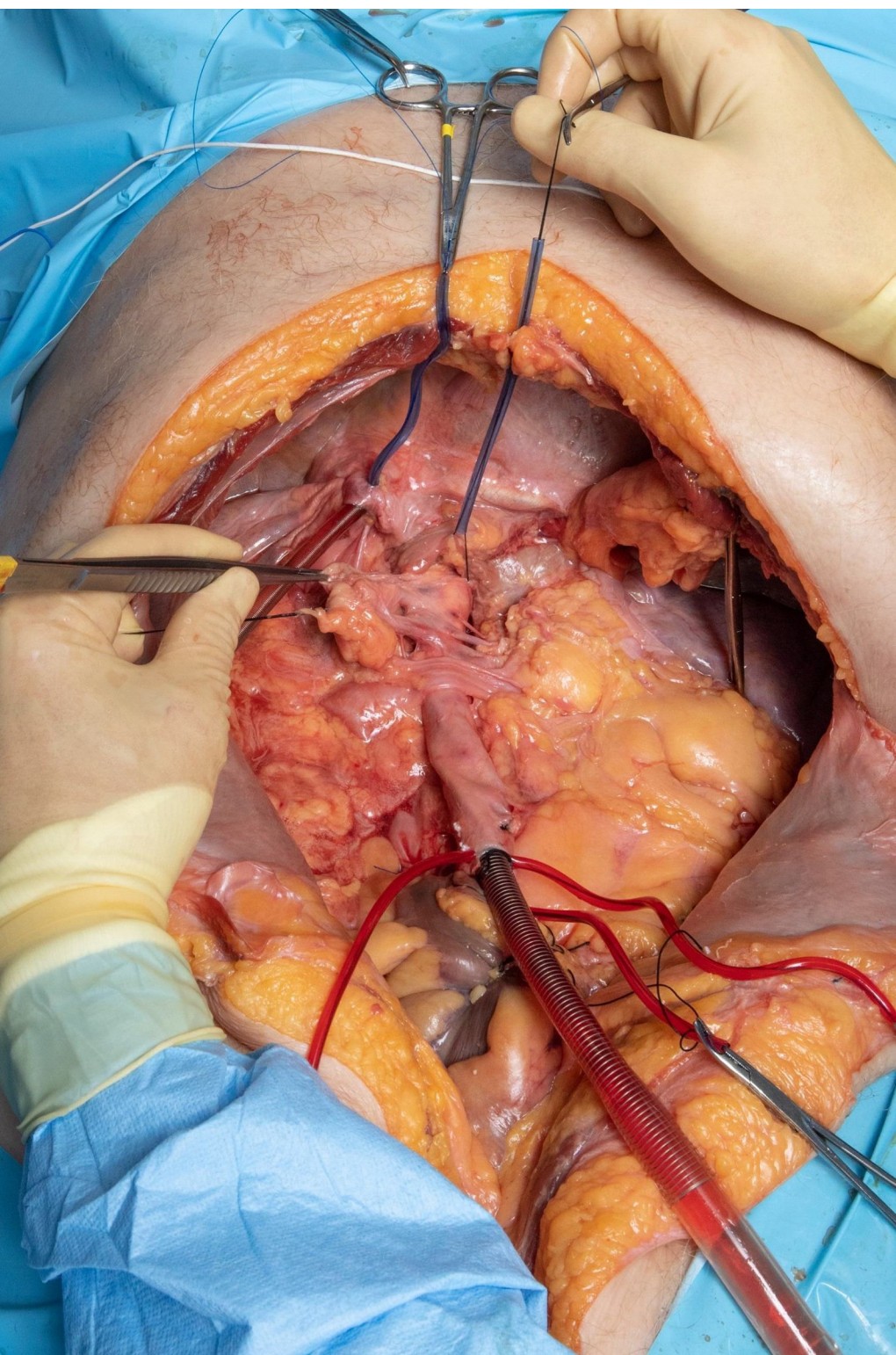

**Fig 3. Second approach with evisceration of the abdomen.** A right and left medial visceral rotation were performed and the entire colon was mobilized. The venous cannula was advanced into the thoracic aorta and numerous branches and tributaries of the aorta and IVC were ligated to reduce systemic flow.

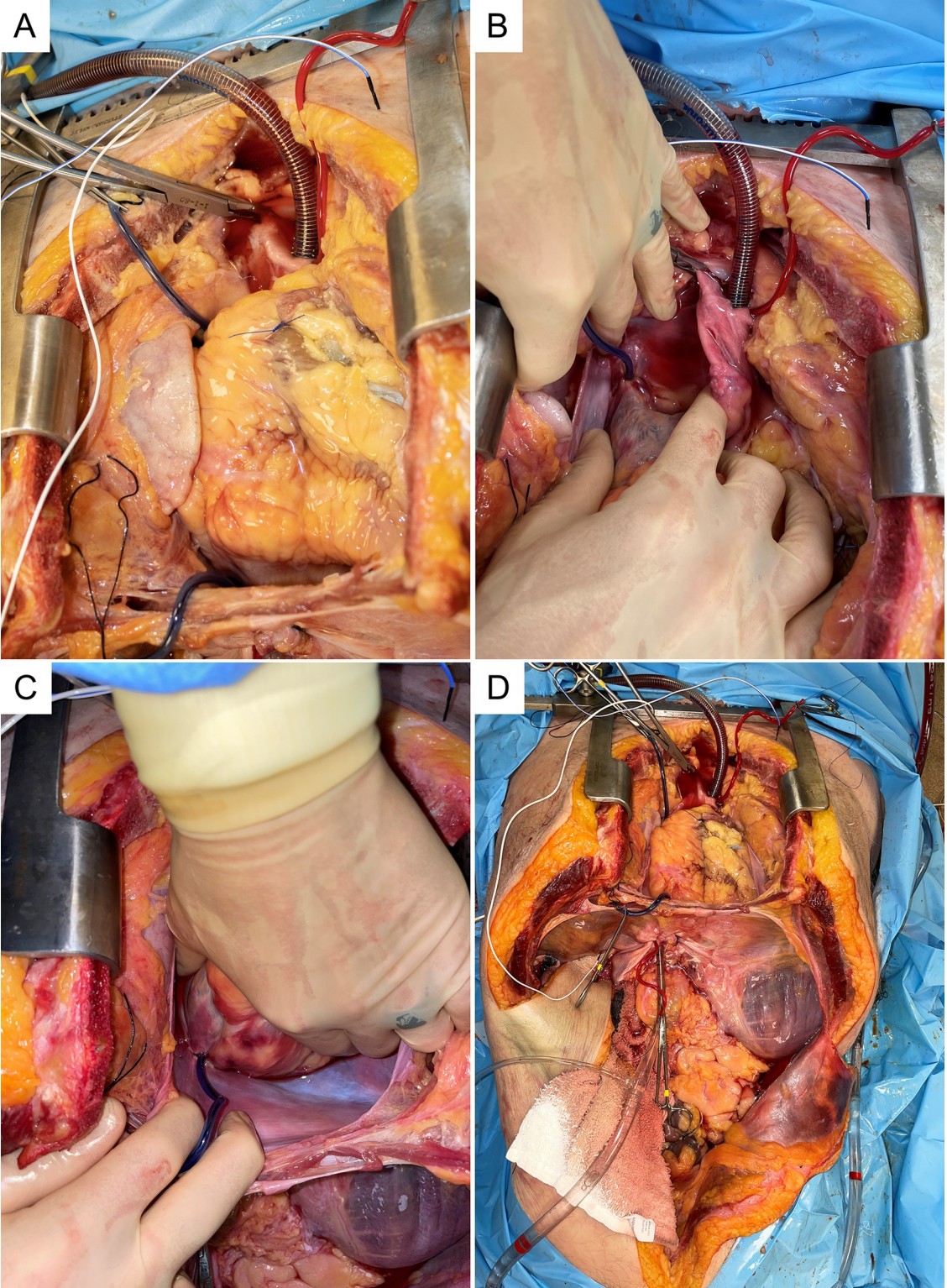

**Fig 4.** A-D Third approach with midline sternotomy. The receiving catheter was moved into distal ascending aorta and impedance measurements were performed using multiple thoracic positions including the right atrium, diaphragm and esophagus.

canula and inserted it into the distal ascending aorta and secured with a purse string suture. The aorta distal to this was then clamped to isolate flow out of the heart to the cannula alone. Flow was again established at 1.5 L/min in the same fashion as before (see S2 Video).

## Measuring venous air embolism through various pathways

In order to demonstrate viability of this cadaveric simulation model, a digital oscilloscope was used to measure impedance across multiple bipolar montages using needle tips in combinations of the skin, esophagus, diaphragm, and pulmonary artery. Impedance waveforms were recorded after delivering 30 to 60 mL boluses of air during each simulation run via independent vascular access in the IVC. At the end of our assessment period, a hole was noted in the RV which most likely occurred due to the high pulmonary artery resistance from the cadaveric lung tissue and a forward pressure regularly and often above 50 mmHg. At the conclusion of simulation, the volume of normal saline infused through all simulations were 46 L.

## Results

In all recording electrode positions in the intrathoracic approach, respiratory variation could be visually qualitatively detected through mechanical ventilation of the cadaveric model. In multiple electrode positions within the surgical field, entrained air could be detected with adequate signal to noise ratio. Fig 5 demonstrates an example recording from a single pair of electrodes in the esophagus and right atrium (to emulate two recording regions which could be accessed via a minimally invasive approach via orogastric tube and vascular introducer, for example). These air injections were readily detectable via the impedance recordings as stepped increases in the intravascular resistance which returned to baseline over the course of minutes. Further experimentation is needed to validate the monitoring apparatus, which is outside of the scope of our current cadaveric model development.

## Discussion

We have developed a cadaveric model to simulate VAE to allow for development of automated detection and monitoring tools. We plan to use this scheme to perform pre-clinical and non-animal based investigation of electrical impedance-based tools given the potential for low cost and the promise of minimally-invasive detection. We speculate that high speed computation analyses may permit identification of low signal-to-noise ratio signals that may not have been previously clinically useful. However, this cadaveric model may prove useful in the development of tools which use other detection modalities as well prior to animal or in vivo human validation. We note that non-invasive evaluation during (human) TEE bubble studies preoperatively or for cardiac diagnostic purposes may also be a useful proving ground for percutaneous detection, in later stages of tool development. While other experimenters have used cadavers in with extracorporeal membrane oxygenation (ECMO) machines with varying success in one study [10], to our knowledge, none have employed the reversed circulation and isolation techniques that we employed in this model. The protocol is summarized in Supplement 3.

In our cadaveric model, we had success with mechanical ventilation and reverse bypass circulation independently, however the combination of both techniques led to massive pulmonary edema and return of fluid through the endotracheal tube and into multiple potential spaces which impaired mechanical ventilation necessitating its discontinuation. We were able to perform simulation with each intervention separately. Also, we note that we are currently injecting air by hand via a separate venous access point, and that this may result in cardiac volume changes that could cause our apparatus to detect signal however the flows from the bypass machine are much greater than these volumes and we do not seen any physical movement

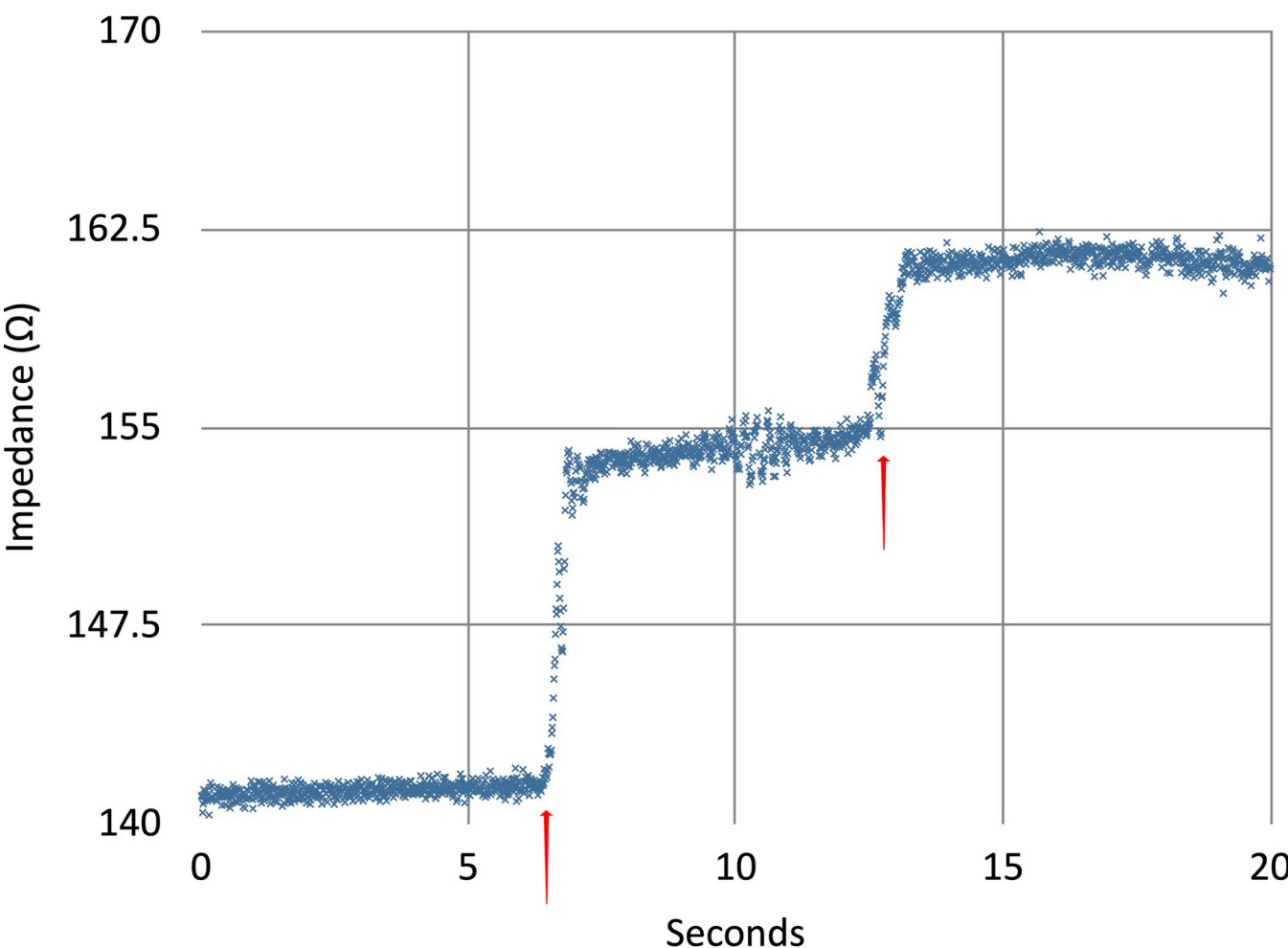

**Fig 5. Bipolar impedance recordings measured between right ventricular to esophagus measured at a frequency of 50 kHz with samples recorded 100 times per second.** The steps at approximately 7 and 13 seconds correlated with injection of 30 mL of air (red triangles) into a central venous line during reversed cardiac bypass circulation with central catheter placement via thoracotomy. Recordings were performed using the Analog Discovery 2 digital oscilloscope (Diligent Corp., Pullman, Washington, USA) and proprietary scripting software.

associated with injection. Lastly, another limitation of our technique is the progressive volume loss despite more central cannulation which places a limit on the duration of effective simulation. We would estimate that if we had performed central cannulation via thoracotomy initially with maximal ligation of peripheral vasculature, we could support 20-30 minutes of simulation using a cadaver based on the cumulative simulation duration in these 3 approaches. It may be reasonable to use albumin instead of crystalloid solution to attempt to allow oncotic pressure to support intravascular volume expansion given evidence of efficacy in one study in ECMO with respect to survival in living patients [11], though evidence overall in resuscitation has been mixed [12–14] and its utility in allowing continued perfusion in cadavers is hard to predict. This scheme also requires non-preserved specimens which have limited stability at thawed temperatures and must be employed promptly. Fortunately, the cadaver remained viable for educational anatomic dissection after our experimentation. In our case, the use of only a single cadaveric specimen is a weakness of this study given that one specimen may not represent the feasibility or difficulty of these approaches in general.

In general, further development in the cadaveric apparatus is needed to improve the fidelity of simulation and aid in detection system development and validation however this model is an effective proving ground for VAE monitoring tool development and validation.

## Supporting information

**S1 Video. Video recording of initial extrathoracic approach with reverse cardiac bypass cardiothoracic perfusion and concurrent mechanical ventilation (0:33).**
(MP4)

**S2 Video. Video of intrathoracic approach with more central reverse cardiac bypass cardiothoracic perfusion, without concurrent mechanical ventilation (1:14).** Note the increase in cardiac volume as perfusion is begun in the first 5 seconds of the recording.
(MP4)

## Acknowledgments

We give special thanks to the cardiovascular perfusionists Edvin Tahay C.C.P. and Michael Bohn C.C.P., and the Mayo Clinic anatomical lab staff for their assistance with this project.

## Author Contributions

**Conceptualization:** Chris Marcellino, Matthew Johnston, Arnoley S. Abcejo.

**Data curation:** Chris Marcellino.

**Funding acquisition:** Chris Marcellino, Arnoley S. Abcejo.

**Investigation:** Nathaniel L. Robinson, Chris Marcellino.

**Methodology:** Nathaniel L. Robinson, Chris Marcellino, Matthew Johnston, Arnoley S. Abcejo.

**Project administration:** Chris Marcellino.

**Supervision:** Chris Marcellino, Arnoley S. Abcejo.

**Writing – original draft:** Nathaniel L. Robinson, Chris Marcellino.

**Writing – review & editing:** Nathaniel L. Robinson, Chris Marcellino, Matthew Johnston, Arnoley S. Abcejo.

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
