## [Decision Letter · Decision Letter 0]

1 Aug 2024

PONE-D-23-40838A Human Cadaveric Model for Venous Air Embolism Detection Tool DevelopmentPLOS ONE

Dear Dr. Marcellino,

Thank you for submitting your manuscript to PLOS ONE. After careful consideration, we feel that it has merit but does not fully meet PLOS ONE’s publication criteria as it currently stands. Therefore, we invite you to submit a revised version of the manuscript that addresses the points raised during the review process.

**ACADEMIC EDITOR: **

**Authors are required to reply all the queries, raised by both the reviewers.**

We look forward to receiving your revised manuscript.

Kind regards,

Priti Chaudhary, M.S.

Academic Editor

PLOS ONE

“This work was supported by the Mayo Clinic Anesthesia Small Grant funding, which is made possible by the Mayo Clinic Center for Translational Science Activities (CTSA) through grant number UL1TR002377 from the National Center for Advancing Translational Sciences (NCATS), a component of the National Institutes of Health (NIH).

Reviewers' comments:

Reviewer's Responses to Questions

**Comments to the Author**

1. Is the manuscript technically sound, and do the data support the conclusions?

Reviewer #1: Partly

Reviewer #2: Yes

2. Has the statistical analysis been performed appropriately and rigorously? 

Reviewer #1: No

Reviewer #2: N/A

3. Have the authors made all data underlying the findings in their manuscript fully available?

Reviewer #1: Yes

Reviewer #2: Yes

4. Is the manuscript presented in an intelligible fashion and written in standard English?

Reviewer #1: Yes

Reviewer #2: Yes

5. Review Comments to the Author

Reviewer #1: the introduction is well constructed highlighting the aim of the work, addition of some statistics regarding prevalence of VAE during surgeries, incidence of resulting disabilities and complication is a plus

the methodology detailed description of the three approaches to establish a simulation method and minimize fluid loss

The results: are the recordings done in several sessions to the conducted model??!!! if not several recordings in different trials for the finalized intrathoracic approach is a plus with comparison between the single pair of electrodes to multiple ones

The discussion clearly enlisted the limitations of the study with recommendation of further studies to validate the model. It needs enrichment with other studies e.g. reviews supporting using colloids instead of saline to support intravascular

volume expansion??!!

Reviewer #2: The manuscript is technically sound and well written. It will help in further understanding of techniques in detection of venous air emboli especially in vivo human. However, find below some minor corrects and suggestions for possible improvement in the quality of the manuscripts.

The word aorta was used in significant component of the manuscript, it is good specify the part of aorta use in each context e.g. abdominal aorta, descending thoracic aorta etc

Small intestine" is the more scientifically precise term, while "small bowel" is commonly used in clinical settings. but "small intestine" is generally preferred in scientific and medical literature

The use of words normal saline is more preferable than the word saline.

The SVC was mentioned before that of Line 154 SVC. Please abbreviate the SVC in the first instance.

Line 181 the ….example.) should be example). The full stop should be after the bracket.

Lines 194 other experimenters … are you referring to more than one study? If yes. Please provide additional references.

In my opinion, the use of one cadaver to develop a model may not be adequate. So this issue of utilizing only one cadaver should discussed in the manuscripts as part of the limitations or recommended for father exploration.

6. PLOS authors have the option to publish the peer review history of their article (what does this mean?). If published, this will include your full peer review and any attached files.

Reviewer #1: No

Reviewer #2: **Yes: **Lawan Hassan Adamu

---

## [Editor Report · Decision Letter 1]

12 Aug 2024

A Human Cadaveric Model for Venous Air Embolism Detection Tool Development

PONE-D-23-40838R1

Dear Dr. Chris Marcellino,

We’re pleased to inform you that your manuscript has been judged scientifically suitable for publication and will be formally accepted for publication once it meets all outstanding technical requirements.

Kind regards,

Priti Chaudhary, M.S.

Academic Editor

PLOS ONE
---

## [Editor Report · Acceptance letter]

14 Aug 2024

PONE-D-23-40838R1 

PLOS ONE

Dear Dr. Marcellino, 

I'm pleased to inform you that your manuscript has been deemed suitable for publication in PLOS ONE. Congratulations! Your manuscript is now being handed over to our production team.

Kind regards, 

on behalf of

Dr. Priti Chaudhary 

Academic Editor

PLOS ONE